# Oxidative Stress-Induced Alterations of Cellular Localization and Expression of Aquaporin 1 Lead to Defected Water Transport upon Peritoneal Fibrosis

**DOI:** 10.3390/biomedicines10040810

**Published:** 2022-03-30

**Authors:** Yu-Syuan Wei, Hui-Ping Cheng, Ching-Ho Wu, Yen-Chen Chang, Ruo-Wei Lin, Yu-Ting Hsu, Yi-Ting Chen, Shuei-Liong Lin, Su-Yi Tsai, Shinn-Chih Wu, Pei-Shiue Tsai

**Affiliations:** 1Graduate Institute of Veterinary Medicine, School of Veterinary Medicine, National Taiwan University, Taipei 10617, Taiwan; f08629016@ntu.edu.tw (Y.-S.W.); r09629010@ntu.edu.tw (H.-P.C.); r07629002@ntu.edu.tw (Y.-T.H.); 2Department of Veterinary Medicine, School of Veterinary Medicine, National Taiwan University, Taipei 10617, Taiwan; chinghowu@ntu.edu.tw (C.-H.W.); yenchenchang@ntu.edu.tw (Y.-C.C.); b04609006@ntu.edu.tw (R.-W.L.); 3Graduate Institute of Veterinary Clinical Science, School of Veterinary Medicine, National Taiwan University, Taipei 10617, Taiwan; 4Graduate Institute of Molecular and Comparative Pathobiology, School of Veterinary Medicine, National Taiwan University, Taipei 10617, Taiwan; 5Department of Internal Medicine, National Taiwan University Hospital, Taipei 10002, Taiwan; chenyiting@ntuh.gov.tw (Y.-T.C.); linsl@ntu.edu.tw (S.-L.L.); 6Department of Integrated Diagnostics & Therapeutics, National Taiwan University Hospital, Taipei 10002, Taiwan; 7Graduate Institute of Physiology, College of Medicine, National Taiwan University, Taipei 10051, Taiwan; 8Research Center for Developmental Biology and Regenerative Medicine, National Taiwan University, Taipei 10617, Taiwan; suyitsai@ntu.edu.tw; 9Department of Life Science, College of Life Science, National Taiwan University, Taipei 10617, Taiwan; 10Department of Animal Science and Technology, College of Bioresources and Agriculture, National Taiwan University, Taipei 10617, Taiwan; scw01@ntu.edu.tw

**Keywords:** oxidative stress, cytoskeleton, aquaporin, porcine, peritoneal dialysis

## Abstract

Being one of the renal replacement therapies, peritoneal dialysis (PD) maintains around 15% of end-stage kidney disease patients’ lives; however, complications such as peritoneal fibrosis and ultrafiltration failure during long-term PD compromise its application. Previously, we established a sodium hypochlorite (NaClO)-induced peritoneal fibrosis porcine model, which helped to bridge the rodent model toward pre-clinical human peritoneal fibrosis research. In this study, the peritoneal equilibration test (PET) was established to evaluate instant functional changes in the peritoneum in the pig model. Similar to observations from long-term PD patients, increasing small solutes transport and loss of sodium sieving were observed. Mechanistic investigation from both in vivo and in vitro data suggested that disruption of cytoskeleton induced by excessive reactive oxygen species defected intracellular transport of aquaporin 1, this likely resulted in the disappearance of sodium sieving upon PET. Functional interference of aquaporin 1 on free water transport would result in PD failure in patients.

## 1. Introduction

Peritoneal dialysis (PD) is one of the renal replacement therapies to maintain patients’ lives who suffered from end-stage kidney disease [1,2]. Under PD, the peritoneum is the main exchange surface for removing excess fluids and metabolic wastes [3]. The functional peritoneum contains a thin membrane composed of the mesothelium, sub-mesothelial zone, and capillaries; however, during long-term PD, morphological characterizations for peritoneal fibrosis, such as loss of mesothelial cells, accumulation of extracellular matrix and angiogenesis were commonly observed [4,5,6]. Besides morphological alterations, functional changes were also detected upon the occurrence of peritoneal fibrosis, these above-mentioned changes eventually led to severe complications, such as encapsulating peritoneal sclerosis (EPS) or ultrafiltration failure with high mortality. To date, there were limited treatments for PF in clinical, and the efficacy of those treatments was far from satisfactory [7,8].

Accumulating evidence shows that besides the lubricating function of the peritoneum to avoid friction of abdominal organs, other functions, such as transport of fluids, mediation of inflammation, and tissue repair processes have also been reported [9,10]. Among the above-mentioned functions, the transportability of solutes and fluids is the most critical factor for the success of PD. To transport solutes and water between plasma and peritoneal cavity, functional integrity of the three barriers, namely endothelium of capillaries, peritoneal interstitial space, and mesothelium are thought to be important. According to previous studies, the endothelium was considered the main rate-limiting barrier among these three barriers. Based on the three-pore model describing the functional characteristics of the endothelium, large pores, small pores, and ultra-small pores on endothelium were used to explain the complex movements of water and various solutes [11,12]. This theoretical model was also used to explain the changes of various parameters upon the peritoneal equilibration test (PET), a common method in clinical to depict overall peritoneum transport characteristics to allow adjustment in the individual prescription of PD [13,14]. In long-term PD patients, increasing small solutes transport, decreasing ultrafiltration volume and sodium sieving are the most commonly detected changes upon PET. All these above-mentioned changes in peritoneum permeability are thought to result from the impairment of free water transport and ultrafiltration abilities due to defected water channel protein, aquaporin 1 (AQP1) on both endothelium and mesothelium upon the occurrence of peritoneum fibrosis [15,16]; moreover, Piccapane et al. showed functional AQP1 was crucial for transmesothelial water transport. Another study also demonstrated that AQP1-containing exosomes present in peritoneal dialysis effluent may potentially serve as a biomarker for the integrity of the peritoneum, and reflect the water permeability status of the peritoneum; more importantly, detection of those AQP1-containing exosomes may be used to predict ultrafiltration failure in PD patients [17,18,19]; however, although the changes shown on PET and impairment of AQP1 were considered as relevant indications to fibrosis and angiogenesis, the underlying mechanism of these functional changes on peritoneum remains to be elucidated [20,21,22,23].

Previously, we established a novel porcine model of peritoneal fibrosis using sodium hypochlorite (NaClO) [24]. In this study, we further establish PET in pigs to allow instant evaluation of the transportability of the peritoneum. Through the establishment of PET on pigs, the progression of peritoneal fibrosis and the efficacy of treatments could therefore be monitored and evaluated immediately in live pigs, informative relationship between morphological changes and functional integrity of the peritoneum can also be connected.

## 2. Materials and Methods

### 2.1. Chemicals, Reagents, Antibodies

Unless stated, chemicals and reagents used in the current study were obtained from Sigma Aldrich (St. Louis, MO, USA). Rabbit polyclonal anti-aquaporin 1 (AQP1, #Ab15080), anti-nicotinamide adenine dinucleotide phosphate oxidase 4 (NOX4, #Ab133303), anti-eukaryotic elongation factor 2 (EEF2, #Ab40812), goat polyclonal anti-alpha smooth muscle actin (αSMA, #Ab21027), mouse monoclonal anti-type I collagen (#Ab6308), anti-Von Willebrand factor (vWF, #Ab6994) and anti-8-hydroxyguanosine (8-OHdG, #Ab62623) were obtained from Abcam (Cambridge, UK). Mouse anti-human Cytokeratin (CK) Clones AE1/AE3 (#M351501) were obtained from Dako/Agilent (Santa Clara, CA, USA). Mouse monoclonal anti-aquaporin 1 (#SC-25287) and mouse anti-extracellular signal-regulated kinase (ERK, #SC1647) were obtained from Santa Cruz Biotechnology Inc. (Dallas, TX, USA). Rabbit monoclonal anti-phospho-p44/42 MAPK (pERK, #4370S) was purchased from Cell Signaling Technology (Danvers, MA, USA). Fluorescent conjugated anti-tubulin (#A322588), anti-phalloidin (#A12380) were obtained from Invitrogen/Life Technology (Carlsbad, CA, USA). All secondary antibodies were purchased from Jackson ImmunoResearch Laboratories Inc. (West Grove, PA, USA).

### 2.2. Establishment of Peritoneal Fibrosis

Animal experiments were approved and carried out under the regulation and permission of the IACUC protocol (NTU-106-EL-00165, NTU-110-EL-00099) at National Taiwan University (Taiwan). Nine five-week-old LYD (mixed breed of Landrace–Yorkshire–Duroc) male piglets free from porcine epidemic diarrhea virus, porcine reproductive and respiratory syndrome virus, and porcine circovirus (tested with PCR, data not shown) were purchased from Tung-Ying Agriculture Inc., Taiwan, and housed in groups in a certified animal facility for one week prior to the experiment. NaClO-induced peritoneal fibrosis was established as earlier described [24]. Pigs were randomly divided into three groups (n = 3 in each experimental group) as control, 0.1% NaClO and 0.1% NaClO*2 group. For control and 0.1% NaClO groups, sterile normal saline without or with 0.1% (*v/v*, 15 mM) NaClO was injected intraperinatally on day 6; as for 0.1% NaClO*2 group, sterile normal saline contained 0.1% NaClO was injected on day 6 and day 9. For all groups, pigs were sacrificed on day 13th to obtain tissue samples for further protein and histological evaluations (Figure 1A,C).

### 2.3. Establishment of Peritoneal Equilibration Test (PET)

Before the induction of peritoneal fibrosis, all piglets underwent surgery for peritoneal dialysis catheter (Argyle™ Pediatric Tenckhoff, #8888414201) insertion on day 0. On the day of surgery, piglets were weighed and sedated by intramuscular injection of xylazine (2.2 mg/kg B.W.) and tiletamine/zolazepam (2 mg/kg B.W.) mixture. General anesthesia was induced by intravenous injection of propofol (0.5 mg/kg B.W.) and maintained by both propofol and inhalation of isoflurane. During the surgery, basic physiological parameters such as breath, heart rate, body temperature, blood oxygen level, blood pressure, end-tidal carbon dioxide (EtCO2), and electrocardiogram (ECG) were monitored and maintained stable. After the surgery, the wounds were examined daily, and postoperative antibiotics and analgesics were given intramuscularly for three days. Baseline PET was performed on pigs for initial evaluation of peritoneal function on day 3, the subsequent evaluation after the induction of peritoneal fibrosis was carried out on day 13 before necropsy. To establish PET in pigs, the human PET procedure suggested by Twardowski [13] was followed with minor adjustments. Before the infusion of dialysate, the absence of ascites was checked on every individual. 80 mL/kg B.W. of pre-warmed 2.5% dialysate (Dianeal 2.5%) was injected through a PD catheter within ten minutes and drained out after a four-hour dwell. Blood and dialysate samples were obtained at 0, 30, 60, 120, 240 min respectively, kept at 4 °C before measurement, and analyzed within 24 h. Creatinine, urea nitrogen, and sodium were measured in both blood and dialysate samples; glucose was only measured in dialysate samples to calculate the dissipation rate of glucose. The creatinine concentration in PD effluent was corrected for glucose interference as suggested by Twardowski [13]. In a follow-up analysis, D/P creatinine (dialysate-to-plasma ratio), D/P urea nitrogen, D/P sodium, and D/D0 glucose (ratio of dialysate glucose at every time points to dialysis glucose at 0 dwell time) were calculated to evaluate the transport characteristics of the peritoneum.

### 2.4. Histological and Quantitative Assessments on Mesothelium Integrity, Tissue Thickening and the Amount of αSMA

After pigs were sacrificed, both parietal and visceral peritoneum were collected and fixed overnight (O/N) in 10% neutral formalin before being processed for paraffin embedding. Ten-μm tissue sections were de-paraffinized and stained with hematoxylin and eosin (H&E) for general morphological and histological evaluations as described earlier [24,25]. For tissue fibrosis analysis, Masson’s trichrome stained sections were quantitatively assessed using CellSens software (Olympus, Tokyo, Japan). To quantify the changes in the peritoneum under NaClO injury, an objective multipoint measurement of the surface thickness was carried out as described earlier [24]. In short, CK was used for mesothelium integrity evaluation. A mosaic tile image for the entire tissue sampled was first generated. The length (in μm) of the total tissue surface and CK^+^ region in length were measured. A ratio between CK^+^ length /total tissue surface length was calculated and expressed in percentage. To quantify the amount of myofibroblast, specific marker protein αSMA was used, total signal intensity (in pixel) was calculated and signals from the blood vessel were manually removed. The quantity of αSMA^+^ signal was expressed as positive pixel/μm^2^ tissue area.

### 2.5. Indirect Immunofluorescence Staining

For indirect immunofluorescent assay (IFA), paraffin-embedded tissue sections were used. Tissue sections were deparaffinized with 100% xylene and rehydrated with 100–80% ethanol. Antigen retrieval was carried out by heating tissue sections in 10 mM citrate buffer (pH 6.0–9.0). Non-specific signals were minimized with 1% BSA for 60 min at RT. Tissue were permeabilized with 0.1% Triton-x 100 for 5 min at 4 °C, cells were permeabilized with 0.5% Triton-x 100 for 10 min at 4 °C. Primary antibody incubation was carried out with O/N incubation at 4 °C. Cell nuclei were counterstained with a mounting medium in the presence of diamidino-2-phenylindole (Abcam, #Ab104139). Negative control was performed by the same staining procedure but omitting the primary antibody. Stained slides were evaluated with Olympus IX83 epifluorescence microscopy and analyzed with CellSens software (Olympus, Tokyo, Japan). Background subtraction was performed identically for all images (including control images).

### 2.6. Cell Culture and Cellular ROS Detection Assay

Human mesothelial cell line (MeT-5A, # CRL-9444™), and human umbilical vein endothelial cell line (EA.hy926, #CRL-2922™) were obtained from ATCC (Manassas, VA, USA). MeT-5A was cultured in Medium 199 (M199, Gibco, NY, USA) supplemented with 5% fetal bovine serum, 3.3 nM epidermal growth factor (EGF) (E9644, Sigma), 400 nM hydrocortisone (H0888, Sigma), 870 nM insulin (91077C, Sigma), and 1% penicillin-streptomycin-amphotericin B (Gibco). EA.hy926 was cultured in Dulbecco’s Modified Eagle Medium (D6429, Sigma) supplemented with 10% fetal bovine serum and 1% penicillin-streptomycin-amphotericin B (Gibco). Both cell lines were cultured at 37 °C in a humidified atmosphere with 5% CO_2_. To investigate NaClO-induced reactive oxygen species [26] production, cellular reactive oxygen species detection assay kit (deep red fluorescence) (Ab 186029, Abcam, UK) was used. Cells were seeded at a density of 4 × 104 cells/90 μL per well in a 96-well plate for 24 h. Various concentrations (0–0.1%; 0–15mM) of NaClO solution was added to the cells, and a time course incubation (15–60 min) was performed to evaluate the time-dependent effects on ROS generation by NaClO. After incubation, the supernatant was removed and 100 μL of ROS deep red working solution containing ROS sensor was added to each well. After further incubation for 30 min at 37 °C, fluorescence intensity reflected the amount of intracellular ROS was measured by a microplate reader (SpectraMax M5, Molecular Devices, Silicon Valley, CA, USA) at Ex/Em = 650/675 nm.

### 2.7. Western Blotting

An equivalent amount of protein extract (μg) was resuspended with an appropriate volume of lithium dodecyl sulfate (LDS) sample buffer (NuPAGE™, Thermo Fisher Scientific) in the presence of a sample reducing agent (NuPAGE™, Thermo Fisher Scientific). Samples were heated in a 100 °C dry bath for 10 min and air-cooled to RT before loading on gels. Bio-Rad Mini-PROTEIN^®^ electrophoresis system was used (Bio-Rad Laboratories Ltd., Hertfordshire, UK) and standard manufactory protocol was followed. Proteins were separated by 10% sodium dodecyl sulfate-polyacrylamide gel (SDS-PAGE, gradient T-Pro EZ Gel Solution, T-Pro Biotechnology, New Taipei County, Taiwan) and wet-blotted onto a Polyvinylidene difluoride (PVDF) membrane (Immobilon-P, Millipore, Burlington, MA, USA). After blocking for one hour with Tri-buffered saline-Triton X100 (5 mM Tris, 250 mM sucrose, pH 7.4 with 0.05% *v*/*v* Tween-20 [TBST], supplemented with 5% milk powder) at RT, blots were incubated with primary antibodies at 4 °C for overnight. After three times washing in TBST, a secondary antibody was subsequently added, and blots were incubated at RT for another hour. After rinsing with TBST, a specific protein signal was visualized by chemiluminescence (PK-NEL122, Blossom Biotechnologies Inc., Taipei, Taiwan) and detected under the ChemiDoc™ XRS+ system (Bio-Rad Laboratories, Hercules, CA, USA). The relative intensity of each band was determined using ImageJ software. When necessary, blots were stripped with stripping buffer (ab270550, Sigma) and re-probed for other proteins of interest.

### 2.8. Statistical Analyses

Statistical analyses were performed using GraphPad Prism (GraphPad Software, San Diego, CA, USA). The statistical significance between groups was accessed by one-way ANOVA followed with Mann–Whitney U test for 2-group comparison. Statistical significances were marked with asterisks and expressed as follow: ** = *p* value < 0.01; * = *p* value < 0.05, n.s.= *p* value > 0.05. Data were expressed as mean ± standard deviation (S.D.).

## 3. Results

### 3.1. Functional Changes of Peritoneum Were Detected upon PET after Two NaClO Administrations

As illustrated in Figure 1A, before the induction of peritoneal fibrosis, a PD catheter was implanted in all pigs (day 0). Three days after catheter implantation, the first PET was performed to measure the baseline transport pattern of different solutes on the peritoneum in a healthy state. One week after the induction of peritoneal fibrosis, a second PET and necropsy were performed to evaluate the functional and histological changes in the peritoneum (Figure 1A). Consistently with our earlier publication [24], when peritoneal fibrosis was induced by a single injection of 0.1% NaClO, significant peritoneal injury and fibrosis were observed under histological examination; however, under PET results, no significant differences can be detected between control (grey lines) and 0.1% NaClO-injured groups (green lines) on values of D/D_0_ glucose, D/P urea and D/P creatinine. Nevertheless, the value of D/P sodium showed a mild increase with a less apparent sodium sieving phenomenon in 0.1% NaClO-injured animals (Figure 1B). Despite this reproducible outcome, the main goal of the current study is to bridge morphological alterations with functional changes of the peritoneum upon PET evaluation, we, therefore, modified the frequency of NaClO injection accordingly (Figure 1C). As shown in Figure 1D, under two times 0.1% NaClO administrations, the value of D/D_o_ glucose was significantly decreased at all time points evaluated (red line), reflecting more rapid dissipation of glucose. In addition, values of D/P urea and D/P creatinine were also significantly increased at all time points (Figure 1D). Alteration of these three parameters indicated a significant increase in small solutes transport rate after 0.1%*2 NaClO injections. For D/P sodium, values from all time points measured were a significant increase when compared with control baseline values, and the phenomenon of sodium sieving was not observed under this injury condition which indicated that free water transport was impaired after 0.1%*2 NaClO injections (Figure 1D).

### 3.2. Thickening of Peritoneum, Loss of Mesothelial Cells and Accumulation of Myofibroblasts Were More Apparent after Two NaClO Administrations

Apart from evaluating the transportability of the peritoneum, quantitative analyses on the degree of peritoneal fibrosis were also performed. Masson’s trichrome and immunofluorescence staining of mesothelial cell marker, Cytokeratin (CK in green) and myofibroblast marker, alpha-smooth muscle actin (α-SMA in red) were carried out to access the level of collagen deposition, the integrity of mesothelium and the amount of myofibroblast on the parietal peritoneum. Based on Masson’s trichrome staining, thickening of the connective tissue (in light blue) was observed at the sub-mesothelial compact zone after two 0.1% NaClO administrations (Figure 1E). In addition, in contrast to intact mesothelium in the control group, apparent loss of mesothelium and infiltration of myofibroblasts at the sub-mesothelial zone were observed in 0.1%*2 NaClO-injured animals, indicating that severe peritoneal fibrosis occurred under this condition (Figure 1E). In agreement with our earlier findings in single 0.1% NaClO administration [24], quantitative analyses showed a significant increase in peritoneal thickness, loss of mesothelium, and infiltration of myofibroblasts in the 0.1%*2 NaClO group. The changes in these parameters were more apparent when compared with a single NaClO injection, suggesting a more severe peritoneal fibrosis occurred in our modified model (Figure 1F).

### 3.3. Increased Oxidative Damages on Both Parietal Mesothelium and Vessel Endothelium in NaClO-Injured Pigs

ROS was involved in the pathogenesis of PD-induced peritoneal fibrosis [27,28,29]. To investigate whether ROS-induced damages also occurred in our NaClO-induced peritoneal fibrosis pig model, 8-hydroxy-2′-deoxyguanosine (8-OHdG), a marker for DNA oxidative damage was used to evaluate the level and pattern of oxidative damages on parietal peritoneum. In contrast to minimal detection of 8-OHdG signal in the control peritoneum, under the injury of 0.1%*2 NaClO, 8-OHdG signals were stronger and more dispersed in all cell types (Figure 2C) when compared to the vehicle control group (Figure 2B); moreover, 8-OHdG signals can be clearly detected in the mesothelium located on the parietal peritoneum (yellow arrowheads, Figure 2B1,C1) as well as on the endothelium of the vessels (white arrowhead, Figure 2C2), these indicated that administration of 0.1%*2 NaClO led to oxidative damages in our pig model.

### 3.4. Peritoneum AQP1 Expression Was Significantly Decreased in NaClO-Injured Pig

The sodium sieving phenomenon upon PET is owing to the initial water transport across the peritoneum. To investigate the underlying mechanism of the disappearance of sodium sieving observed in Figure 1D, cellular localization and total protein expression of AQP1 on the parietal peritoneum were examined. AQP1 is a water channel protein that belongs to the ultra-small pore on the peritoneum and was considered to be essential for sodium sieving [12,15,16,30]. Consistently with observation in humans [31,32], immunofluorescence staining showed a distinct AQP1 signal on the apical region of the mesothelium (Figure 3A1), marked with arrowheads) and on the adluminal area of the endothelium (Figure 3A2), double stained with vWF, marked with arrowheads). In sharp contrast, under injury of 0.1%*2 NaClO, signals of AQP1 on both mesothelium and endothelium were weakened and scattered into the cytosol (Figure 3A3,A4) indicated 0.1%*2 NaClO treatment altered the cellular distribution of AQP1. When total protein expression of AQP1 on the peritoneum was examined, a significant decrease (~80%) of AQP1 protein expression was detected after NaClO-induced injury (Figure 3B, eukaryotic elongation factor 2 (EEF2) was used as protein loading control). Taken together, NaClO significantly affected both cellular localization and total protein expression of AQP1.

### 3.5. In Vitro Investigation of Potential Signaling Pathways for NaClO-Induced Oxidative Stress

To investigate the potential signaling pathways that were activated upon NaClO administration, we performed sets of in vitro experiments using both mesothelial (MeT5A) and endothelial (EA.hy926) cell lines. We found that both mesothelial and endothelial cell lines treated with 0.01% NaClO were not affected by cell viability when (3-(4,5-Dimethylthiazol-2-yl)-2,5-diphenyltetrazolium bromide (MTT) assay was performed (Appendix A); however, when MeT5A was treated with 0.01% NaClO, a significant increase of ROS production could be observed after 30 and 60 min (Figure 4A), this NaClO-induced excessive ROS production was further confirmed by the fact that addition of an antioxidant, vitamin E significantly reduced intracellular ROS in MeT5A (Appendix A). Based on immunofluorescence results, oxidative stress markers, 8-OHdG and NOX4 were significantly increased on MeT5A after incubation of NaClO for 30 min (Figure 4B). Interestingly, when we further examined which ROS-induced signaling pathway was involved, a NOX4-associated signaling pathway was detected. We observed an early increase of the NOX4 protein expression after 1-h NaClO co-incubation with subsequent activation of the ERK pathway as increased expression of phosphorylated ERK was detected at a later stage (Figure 4C). Similar to mesothelial cells, a significant increase in ROS production can be measured on EA.hy926 after 0.05% NaClO co-incubation (Figure 4D); moreover, oxidative stress markers, 8-OHdG and NOX4 were also detected on EA.hy926 after 30 min (Figure 4E). In addition, similar to MeT5A, activation of NOX4 and ERK pathway was also observed on EA.hy926 as the significant conversion of ERK to phosphorylated ERK was detected (Figure 4F); however, different from MeT5A, the activation of ERK pathway was shown earlier on EA.hy926 after 3 h incubation of NaClO. These data suggested that despite NaClO-induced oxidative damages in both mesothelium and endothelium, the sensitivity or the tolerance of different cells to intracellular ROS may vary.

### 3.6. Disruption of Cytoskeleton and Alteration of Cellular Localization of AQP1 Were Observed under NaClO Co-Incubation

Earlier publications showed excessive ROS would affect cytoskeleton structure [33]. We observed in the current study, an apparent cell shrinkage after MeT5A and EA.hy926 were treated with NaClO (Figure 5); moreover, in contrast to well-organized cytoskeleton in control cells, disorganized and disappearance of filamentous F-actin (in red) and tubulin (in green) were observed. Besides morphological changes in cell size, shape and cytoskeletal structure, we also detected punctate accumulation of cytoskeleton remnants in MeT5A (marked in arrowheads) after NaClO-induced injury (Figure 5A). Similar to MeT5A, under 0.05% NaClO, EA.hy926 also showed obvious shrinkage (Figure 5B). Unlike distinct filamentous signals of F-actin in control cells, F-actin signals on EA.hy926 seemed more dispersed after injury (Figure 5B). As for tubulin, instead of clear and stretching nucleus-to-membrane direction in control cells, a blurred, disorganized and fragmented tubulin was observed in NaClO-treated EA.hy926; moreover, pronounced ring-shaped membrane ruffling can also be observed (Figure 5B, green arrowheads). It is known that a disorganized cytoskeleton could affect protein translocation, we, therefore, examined the cellular location and protein expression of AQP1 on NaClO-treated MeT5A and EA.hy926. In contrast to the significant decrease in AQP1 from in vivo data, we observed no changes in total protein expression in both MeT5A and EA.hy926 (Figure 6A,D); however, unlike typical peri-nucleus and cell membrane distribution of AQP1, the aggregated signal was detected around the nucleus region after injury of NaClO (Figure 6B,E). A self-defined quantitative analysis measuring membrane distribution of AQP1 (Appendix A) demonstrated that a clear centralization (aggregation) rather than dispersed membrane distribution of AQP1 was detected in both MeT5A and EA.hy926 after NaClO co-incubation (Figure 6C,F).

## 4. Discussion

Morphological and functional alterations on the peritoneum upon the occurrence of peritoneal fibrosis were often depicted in long-term PD patients, these changes resulted in the inadequacy of dialysis [23,34] and were considered the main cause of technical failure in PD [35]; therefore, better knowledge of the underlying mechanism of peritoneal fibrosis is necessary to improve the efficiency of PD and the survival of patients. In this study, we successfully established PET to monitor the instant transport characteristics of the peritoneum on live pigs. The success of this approach in pigs created a new possibility to monitor continuously functional changes of the peritoneum along with the progression of peritoneal fibrosis and the recovery process upon treatments. In addition, a limited quantity of biological samples for further analysis and rapid absorption of dwelling dialysate that existed when performing PET in rodent species could also be overcome by our porcine model. Through comprehensive functional and histological evaluations, we demonstrated that histological changes may not fully reflect on the functional integrity of the peritoneum; thus, functional evaluation using PET in combination with histological assessments may provide advanced and informative knowledge upon anti-fibrosis compound screening when apply to pre-clinical trials.

In the clinical context, the adjustments for PD prescriptions are usually based on functional changes in the peritoneum referred from PET rather than the histological changes on the tissue because of its noninvasiveness [3]. Normally, to evaluate the adequacy of PD, ultrafiltration volume, transport of small solutes, such as D/P creatinine and D/P glucose were the most commonly used parameters. In most studies, increased D/P creatinine and decreased ultrafiltration volume were observed under long-term PD [36,37], and these changes were highly associated with disease progression, prognosis, and patient survival rate [38]. Increased density of perfused capillaries on the peritoneum caused by angiogenesis and local production of inflammation cytokines, such as IL-6, TNF-α, IL-1 and IFN-γ were considered as two main causes of the increasing transport of small solutes [39,40,41,42]; besides, the rapid dissipation of glucose led to an ultimate decrease of ultrafiltration volume [43,44]. In our pig model, severe peritoneal fibrosis was observed under histological examination in both single and two administrations of NaClO-injured animals; however, increased transport of small solutes was only detected after two NaClO administrations, suggesting the existence of an information gap in the peritoneum between histological changes and functional integrity. Another crucial parameter in PET is D/P sodium. Sodium sieving, a phenomenon presented by a decrease of D/P sodium at one hour upon PET reflects the free water transportability of the peritoneum during PD [45,46]. One earlier study showed that long-term PD patients with ultrafiltration failure exhibited decreased dip in sodium sieving [47]. Apart from this, decreased or loss of sodium sieving was considered to be an early index for predicting the occurrence of EPS [48,49]. Despite the mechanistic reason behind the decreased or the loss of sodium sieving during long-term PD reminds unclear, AQP1 was thought to be part of the picture [12].

AQP1 is the most abundant aquaporin presented in the human peritoneum [31]. In AQP1 knockout mice, sodium sieving disappeared, strongly supporting the relevance of AQP1 with ultrafiltration in PD [15]. Recent studies from PD patients also showed that variant AQP1gene expressions were associated with distinct transport characteristics with different risk levels of ultrafiltration failure [18]. Despite accumulating, evidence indicating the importance of AQP1 upon PD, changes in AQP1 during the progression of peritoneal fibrosis remain unclear. In this study, besides a decrease in total AQP1 protein expression, disruption of cellular localization upon NaClO injury was observed. Since AQP1 requires membrane localization to take its effects, dis-localization of AQP1 will lead to dysfunction of AQP1 and this was likely due to disruption of cytoskeleton observed in our in vitro study. Firstly, NaClO resulted in excessive ROS, accumulation of intracellular ROS led to increased NOX4 and phosphorylation of ERK, activation of this signaling pathway was considered to play an important role in enforcing the positive feedback loop in fibrosis [50]. Earlier studies showed that although physiological ROS took part in the proper dynamic of the cytoskeleton, excessive ROS would impairs cytoskeleton polymerization [33]. In this study, we observed disorganized tubulin and F-actin in both mesothelial cells and endothelial cells under NaClO-induced oxidative damage. Disrupted F-actin partly explained the morphological changes of the cells, the disorganized tubulin likely interfered with intracellular transport of AQP1 from the endoplasmic reticulum to the cell membrane and resulted in the loss of sodium sieving. Secondly, the thickening of the sub-mesothelium compact zone with accumulated collagen may also contribute to the impaired water transport [45]; nevertheless, in the current study, we could not exclude the possibility that both water and sodium transportability was compromised.

Earlier in vivo and in vitro studies revealed that oxidative stress played a significant role in inducing progressive injury on the peritoneum which subsequently developed into peritoneal fibrosis and ultrafiltration failure [28,29,51,52]. Both excessive production of ROS and impairment of the antioxidant defense system resulted in exorbitant oxidative stress and initiated severe complications. Under the stimulation of PD solutions, glucose degradation product (GDP), advanced glycan end-products (AGEs-RAGE) interaction, and acid pH increased the production of ROS and promoted inflammation, fibrosis, and angiogenesis in the peritoneum [26,27,53,54]. In vitro studies showed that human mesothelial cells incubated under a high glucose environment, ROS, chemotactic peptide-1 (MCP-1), vascular endothelial growth factor (VEGF), transforming growth factor-beta (TGF-β), and fibronectin were all increased [27,55]. In our NaClO-induced peritoneal fibrosis pig model, we previously reported significant peritoneal fibrosis and overexpression of fibrosis-related factors, such as CX3CL1 and TGFβ on the peritoneum. In the current study, our in vitro and in vivo data further demonstrated that excessive production of ROS and oxidative damages were detected in both mesothelium and endothelium. Although mesothelial cells were at the frontline for NaClO-associated oxidative damages, our data indicated that not only mesothelial cells, but also other types of cells within the peritoneum experienced ROS damages.

For NaClO-induced oxidative stress on the peritoneum, both primary and secondary effects might be involved. Primary ROS production likely came from direct NaClO to hypochlorous acid (HClO) degradation [56,57], and secondary oxidative stress may be due to the NOX4-associated loop that further exaggerates ROS production and damages. NOX4 belongs to the NADPH oxidases (NOX) family that transfer electrons across biological membranes and generate ROS [58]. Among different isoforms of NOXes, NOX4 was found to participate in the fibrotic process of various organs. NOX4-derived ROS was thought to induce by TGFβ, and the produced ROS would facilitate the release of the active form of TGFβ, this subsequently formed a positive feedback loop and aggravated fibrosis. Despite earlier studies showing that inhibition or knockout of NOX4, fibrosis on lung, skin, liver, and kidney was alleviated [50]. Scarcely information is available on the role of NOX4 in peritoneal fibrosis. A study published in 2019 showed that when human peritoneal mesothelial cells were incubated with TGF-β1, both mRNA and protein expression of NOX4 was upregulated [59]. In agreement with this study, we showed both mesothelial cells and endothelial cells expressed stronger NOX4 signal after NaClO incubation; however, only mesothelial cells showed significant upregulation in protein expression of NOX4 in western blot. Besides upregulation of NOX4, downstream signaling pathway, phosphorylation of ERK was also activated in both types of cells. One interesting finding was that we observed significant differences between the amount of cellular ROS generated in mesothelial cells and endothelial cells under the same stimulation, which reflected that mesothelial cells were more sensitive to ROS damages than endothelial cells; moreover, we also observed that under NaClO treatment, endothelial cells showed a faster response to ROS stimulation when compared with mesothelial cells, this was evidenced by earlier detection of ERK to phospho-ERK conversion (peak conversion at 24 h and 3 h for mesothelial cells and endothelial cells, respectively) in endothelial cells. These above-mentioned sensitivities and cellular response differences between cell types may partially explain the inconsistency in either morphology or protein expression as well as intracellular ROS production and its subsequent disruption of cytoskeleton between mesothelial cells and endothelial cells.

## 5. Conclusions

We showed that accumulation of intracellular ROS-induced NOX4-associated oxidative damages on both mesothelium and endothelium, which subsequently disrupted cytoskeleton structure and affected intracellular AQP1 translocation and protein expression. Dysfunction in the amount and cellular localization of AQP1 interfered with water transport upon PET and resulted in the disappearance of sodium sieving. A summarized hypothetical model is illustrated in Figure 7. The above-mentioned consequences compromised both morphological and functional integrity of the peritoneum and resulted in PD failure in patients.

## Figures and Tables

**Figure 1 biomedicines-10-00810-f001:**
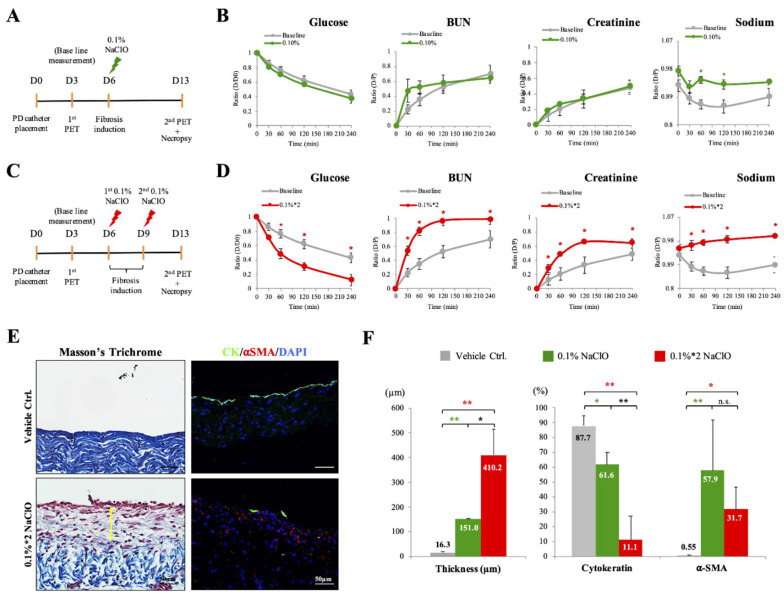
Histological and functional evaluations on peritoneum in NaClO-injured pig models. (**A**) Experimental schedule of single NaClO injection-induced peritoneal fibrosis. (**B**) PET results of a single 0.1% NaClO injury model showed minor differences in D/P sodium but no difference in D/P glucose, D/P urea, and D/P creatinine after a single 0.1% NaClO injection when compared to baseline PET (in grey). (**C**) Experimental schedule of twice NaClO injections-induced peritoneal fibrosis. (**D**) PET results of twice 0.1% NaClO injury model showed significant differences in all parameters measured, D/P glucose, D/P urea, D/P creatinine, and D/P sodium values measured in NaClO injury animals were significantly different from baseline values. (**E**) Significant thickening (yellow double arrow) of parietal peritoneum was observed on Masson’s trichrome staining after two injections of 0.1% NaClO. Fragmentation of mesothelium (labeled with CK in green) and infiltration of myofibroblasts (labeled with α-SMA in red) was also observed. (**F**) Quantitative analyses showed significant thickening of the peritoneum, loss of mesothelial cells, and accumulation of myofibroblasts under both injury models by 0.1% NaClO and 0.1%*2 NaClO. Representative images from 3 individual pigs of each group were presented. Asterisks indicated significant differences between groups (* *p* < 0.05, ** *p* < 0.01, N.S. not statistical different).

**Figure 2 biomedicines-10-00810-f002:**
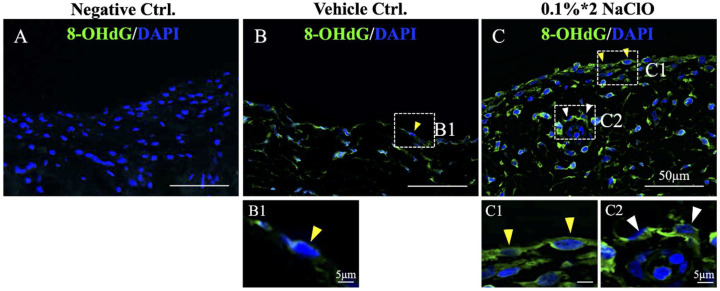
Detection of oxidative damages on both mesothelium and vessel endothelium in NaClO-injured pigs. (**A**) Primary antibody of 8-OHdG was not added for negative control. (**B**,**C**) After 0.1%*2 NaClO injury, signal of 8-OHdG, a marker for DNA oxidative damage became significantly stronger and distributed across all kinds of cells on the parietal peritoneum. (**B1**) Enlarged image depicted a minimal expression of 8-OHdG on mesothelial cells (yellow arrowheads) in control pigs; a significant increase in 8-OHdG signals were detected on both mesothelium ((**C1**), yellow arrowhead) and endothelial cells ((**C2**), white arrowhead) that surrounded vessels. For each experimental condition, 10 images from 3 different pigs were evaluated and representative images were presented.

**Figure 3 biomedicines-10-00810-f003:**
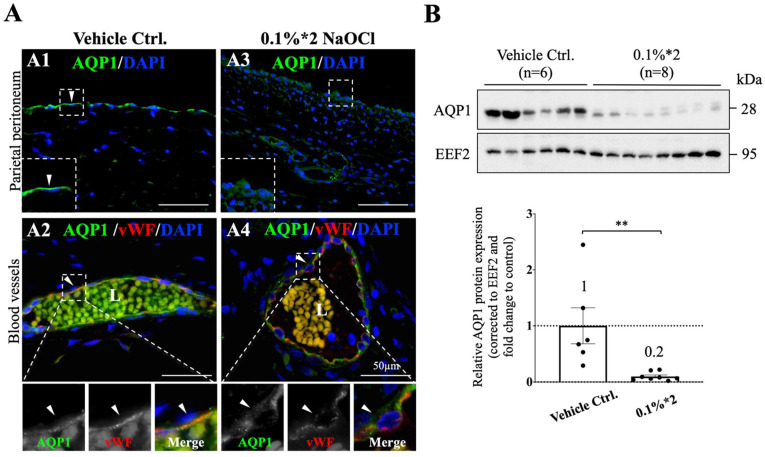
Cellular localization and protein expression of peritoneum AQP1 was altered in NaClO-injured pigs. (**A**) Immunofluorescent staining showed that AQP1 appeared on both mesothelial cells (**A1**) and endothelial cells (**A2**) that surrounded vessels. AQP1 was double stained with vWF to confirm the identity of an endothelial cell. In the vehicle control group, AQP1 mainly appeared on the apical and adluminal region of cells (white arrowheads); however, after 0.1%*2 NaClO injury, signals of AQP1 on both types of cells became weakened (**A3**) and more scattered into the cytosol (**A4**). (**B**) Quantitative analyses on whole peritoneal tissue homogenates showed that total protein expression of AQP1 on parietal peritoneum was significantly decreased (80% less) in 0.1%*2 NaClO-injured pigs. Eukaryotic elongation factor 2 (EEF2) was used as protein loading control, AQP1 signal was first corrected with loading control EEF2 before fold change comparisons to control animals. L lumen of vessels. Representative images from 3–4 individual pigs of each group were presented. For tissue sampling, 2 samples from different regions of the peritoneum were taken from every individual for further analysis. Whole membrane images of western blotting could be found in Appendix A. Asterisks indicate significant differences between groups (** *p* < 0.01).

**Figure 4 biomedicines-10-00810-f004:**
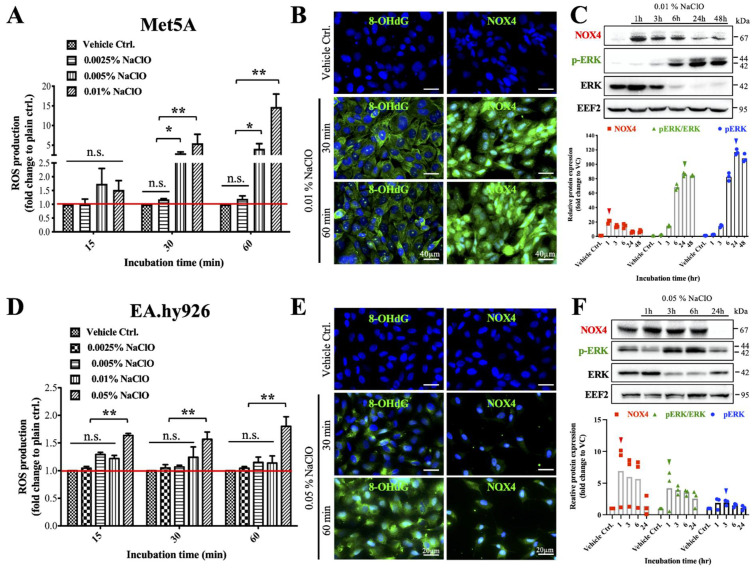
NaClO-induced NOX4-associated oxidative damages on both mesothelial and endothelial cells. (**A**) Significant cellular ROS production was detected when MeT5A was treated with 0.005% and 0.01% NaClO after 30 and 60 min. (**B**) Immunofluorescence results showed MeT5A with increased signals of 8-OHdG and NOX4 after being treated with 0.01% NaClO. (**C**) Time-dependent incr in NOX4 protein expression of ease was observed after 1 h. Subsequent activation of the ERK pathway as evidenced by increased expression of phosphorylated ERK after three hours. Arrowheads indicated peak protein expression. (**D**) Significant cellular ROS production was detected when EA.hy926 was treated by 0.05% NaClO after 15, 30 and 60 min. (**E**) Immunofluorescence results showed increased 8-OHdG and NOX4 signal intensity as well as round morphology of EA.hy926 after being treated with 0.05% NaClO. (**F**) Similar to MeT5A, time-dependent increases in ROS-related protein expressions on EA.hy926 under 0.05% NaClO were quantified. A peak of NOX4 protein expression was observed after 1 h. Then, activation of the ERK pathway was demonstrated by increased expression of phosphorylated ERK after 3 h; however, the protein expression of phosphorylated ERK went back to normal after 24 h. Peak protein expression was labeled in arrowheads. Triplet experiments were performed for cellular ROS production assay. For each experimental condition, 10 images were evaluated, and representative images were presented. Whole membrane images of western blotting could be found in Appendix A. Asterisks indicate significant differences between groups (* *p* < 0.05, ** *p* < 0.01, N.S. not statistical different).

**Figure 5 biomedicines-10-00810-f005:**
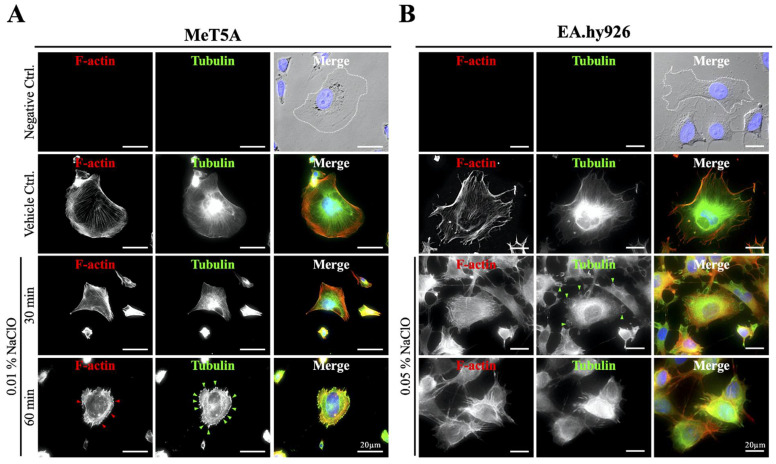
Changes of the cytoskeleton on mesothelial and endothelial cells under NaClO-induced oxidative damages. Double staining of F-actin (labeled in red) and tubulin (labeled in green) was performed on MeT5A and EA.hy926 to observe the changes in cytoskeletal under NaClO stimulation. (**A**) MeT5A showed obvious shrinkage and disorganized patterns of F-actin and tubulin under 0.01%NaClO after 30 and 60 min. Also, punctate accumulation of cytoskeleton (red and green arrowheads) was observed at the edges of the cells. (**B**) Under 0.05% NaClO, EA.hy926 showed a dispersed signal of F-actin and blurred tubulin signal after 30 and 60 min. Besides, ring-shaped membrane ruffling (green arrowheads) was observed after co-incubation of 0.05% NaClO. For each experimental condition, 10 image frames were evaluated, and representative images were presented.

**Figure 6 biomedicines-10-00810-f006:**
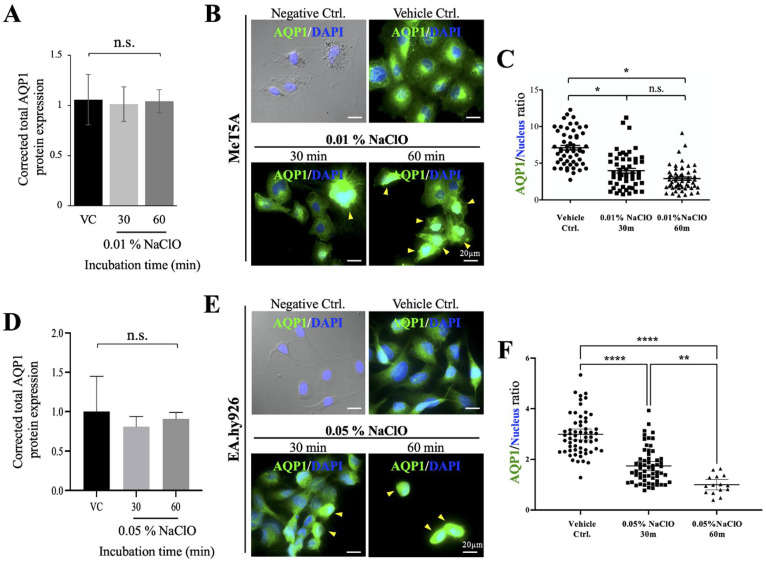
Cellular localization and protein expression of AQP1 in MeT5A and EA.hy926 under NaClO-induced oxidative damages. (**A**) Total AQP1 protein expression of MeT5A showed no differences after being treated with 0.01% NaClO within 60 min. (**B**) Immunofluorescence staining of AQP1 in MeT5A showed an aggregated signal (yellow arrowheads) around the nucleus region under 0.01% NaClO after 30 and 60 min when compared to a dispersed membrane pattern in control cells. (**C**) Self-defined cellular AQP1 distribution analyses showed significant differences in AQP1 distribution after MeT5A was treated with 0.01% NaClO after 30 and 60 min. (**D**) Total AQP1 protein expression of EA.hy926 showed no differences after being treated with 0.05% NaClO within 60 min. (**E**) Immunofluorescence staining of AQP1 in EA.hy926 showed aggregated signals (yellow arrowheads) around the nucleus region under 0.05% NaClO after 30 and 60 min when compared to control cells. (**F**) Quantification analyses showed the more dispersed cellular distribution of AQP1 in control EA.hy926 cells, and a concentrated (aggregated) AQP1 distribution after treatment with 0.05% NaClO. For each experimental condition, except for EA.hy926 at 60 min, 50 cells were evaluated, and representative images were presented. (* *p* < 0.05, ** *p* < 0.01, **** *p* < 0.0001, N.S. not statistical different).

**Figure 7 biomedicines-10-00810-f007:**
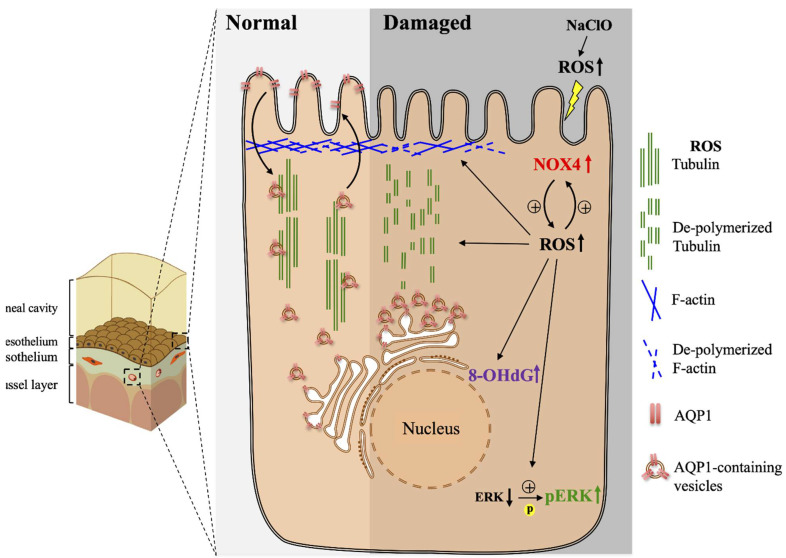
Summarized model for oxidative stress-induced disruption on the cellular transport of AQP1. Upon NaClO stimulations, direct and indirect production of excessive ROS activated NOX4 associated signaling pathways and promoted phosphorylation of ERK, which subsequently enforced a positive feedback loop in producing ROS. Accumulation of ROS led to oxidative stress and enhanced depolymerization of the cytoskeleton. Disruption of both actin and tubulin interfered with intracellular transport of AQP1 from the endoplasmic reticulum to the cell membrane. Failure for AQP1 to reach the cell membrane resulted in the impairment of water transport, which led to the loss of sodium sieving on PET.

## Data Availability

Protocols and materials used in current study, as well as initial raw data collected, are available for the corresponding author on reasonable request.

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
