# Peer review of "Oxidative Stress-Induced Alterations of Cellular Localization and Expression of Aquaporin 1 Lead to Defected Water Transport upon Peritoneal Fibrosis"

_biomedicines, 2022, doi:10.3390/biomedicines10040810_

Round 1

Reviewer 1 Report

Wei et al. report that oxidative stress might have essential roles in peritoneal fibrosis through altering aquaporin(AQP)1 expression. This work is scientifically sound, and the results are convincing. I have several questions about this work.

  1. In figure 3, AQP1 expression was shown in the parietal peritoneum and blood vessels. Figure 3 A2 show that AQP1 expression was endothelial cells. So, for confirmation, a double stain for AQP1 and endothelial cell markers might help to understand the expression pattern of AOQ1 in the peritoneum.
  2. Why do different expression patterns in NOX4 in mesothelial and endothelial cells? However, in Figure 4E, immunocytochemistry data shows NOX4 expression was detected in endothelial cells. There are controversial results, which should address more adequately in the result or discussion sections.
  3. In the PET results, how about dialysate effluent volume changes between time 0 and 240 min? If available the data for effluent volume, please add the PET results.
  4. ROS might have important roles in peritoneal fibrosis. Modifying NOX4 by siRNA or inhibitor studies in vitro experiments may be needed to confirm this result.
  5. The quality of the pictures should be improved.

Reviewer 2 Report

Comments to the authors:

In this study, the authors evaluated the role of the oxidative stress on the expression of AQP1 that is responsible of the impairment of water transport upon peritoneal fibrosis. I have some concerns that you can read below.

  • I strongly suggest to improve the introduction section about the importance of AQP1 in peritoneum and PD since the goal of this work is understanding the relationship between its localization and water transport defect upon peritoneal fibrosis.
  • In section 2.4 please indicate the software used to analyze the images.
  • In section 2.5 specify the type of triton used in the experiments.
  • Please insert the catalog number of Abcam kit used to ROS detection.
  • Insert details about the instrument used to measure the fluorescence of ROS assay.
  • I don’t understand the choice to have inserted the figure 1 in the methods section, anyway I suggest to move it in the results section and make it bigger to improve the visualization.
  • Insert description of EEF2, showed in fig.3B, in 3.4 section and in the figure legend.
  • The figure legend 3B is not clear, what is the type of samples loaded, peritoneum and/or vessels? (L lumen of vessels, line 7). Generally, in western blotting experiments on tissue lysates the AQP1 appears as 2 bands, non-glycosylated and mature glycosylated form. Why did the authors show only one? In the S1 figure there are 2 bands and probably the upper band represent mature form, unfortunately the absence of marker profile in total membrane doesn’t clarify this aspect. I suggest to review this result. Why did the authors use EEF2? If they use it as loaded control they should normalize the AQP1 intensity for EEF2 abundance, please clarify it. In S1 AQP1 at 28 kDa appears as double bands with different intensity in ctr and treated pigs, why?

I suggest to show the loading of only one region of peritoneum for each pig, the animals are 3 for groups, and accordingly change the statistical analysis.

  • The quality of figure 4 is too low, please change all images. In panel A I recommend to introduce a gap in the y axis to show the results clearly. I disagree with the choice of graph type shown in E and F figure, different time points are different wells so I think it is more correct to use a column bar graph.

Again specify the use of EEF2.

Why is the cell lines confluence so different in immnunofluorescence experiments (B-E)?

How do the authors explain the difference between two cell lines in ROS production (in Met5A 10 fold bigger than in EA.hy926)? I suppose that already 1 hour of 0.05 % NaOCl treatment induce significant ROS production in endothelial cells because cells are few so they don’t mimic epithelium, see E.

  • The bars plots in figure 6 are too small and illegible. I strongly suggest to perform also western blotting experiments to quantify the AQP1 abundance in both cell lines.
  • Please discuss in the text the reason why the number of data of endothelial cells treated for 60 min with 0.05%NaOCl is smaller than other conditions.
  • Can the authors perform water transport experiments, for exemple with calcein, in the same conditions reported in figures 4 and 5?.
  • There are other important studies about the role of AQP1 in the peritoneum and its involvement in

the PD, the authors should insert citations of PMID: 30970608, PMID: 34830416.

Supplemental materials

  • Specify the method used for the assessment of cell viability.
  • I don’t understand the reason why for control condition there was not error bar in all figures. Please change all plots according this observation.
  • Please indicate the samples loading on the western blotting original images.

Round 2

Reviewer 1 Report

The authors adequately address my comments.

Before accepting this manuscript, the authors need to check again the term "NaOCl or NaClO" in all figures. 

Reviewer 2 Report

The manunscript can be accepted in this form.

Author Response

Thank you very much for your comment